# Text Mining and Topic Analysis for Ostriches’ Welfare Based on Systematic Literature Review from 1983 to 2023

**DOI:** 10.3390/vetsci11100477

**Published:** 2024-10-05

**Authors:** Annalisa Previti, Vito Biondi, Mehmet Erman Or, Bengü Bilgiç, Michela Pugliese, Annamaria Passantino

**Affiliations:** 1Department of Veterinary Sciences, University of Messina, 98168 Messina, Italy; annalisa.previti@yahoo.it (A.P.); vito.biondi@unime.it (V.B.); annamaria.passantino@unime.it (A.P.); 2Department of Internal Medicine, Faculty of Veterinary Medicine, Istanbul University-Cerrahpasa, 34098 Istanbul, Turkey; ermanor@iuc.edu.tr (M.E.O.); bengu.bilgic@iuc.edu.tr (B.B.)

**Keywords:** ostriches, welfare, machine learning, research, husbandry, text mining, topic analysis

## Abstract

**Simple Summary:**

The present study aimed to provide critical analysis in a systematic review of the peer-reviewed literature indexed in the Scopus database to analyze the studies on ostriches’ welfare in the last forty years in order to identify knowledge gaps in the literature. A bibliometric analysis of metadata and content analysis based on text mining and topic modeling techniques has been conducted on a sample of approximately n. 987 publications, confirming the increased focus on ostriches’ welfare.

**Abstract:**

Ostriches can be utilized as multipurpose animals suitable for producing meat, eggs, feathers, and leather. This growing interest in ostrich farming leads to an increased demand for comprehensive information on their management. But, little attention is paid to the consequences for their welfare. The study aimed to perform a research literature analysis on ostriches’ welfare using the text mining (TM) and topic analysis (TA) methods. It identifies prevailing topics, summarizes their temporal trend within the last forty years, and highlights potential research gaps. According to PRISMA guidelines, a literature exploration was achieved using the Scopus^®^ database, retaining keywords about ostriches’ welfare. Papers distributed in the English language from 1983 to 2023 were included. Descriptive statistics, TM, and TA were applied to a total of n. 122 documents included. The findings revealed an increasing trend in research records since 1994. TM recognized the terms with the highest weighted frequency and TA identified the main topics of the research area, in the following order: “health and management”, “feeding and nutrition”, “welfare reproduction”, “egg production”, and “welfare during transport”. The study confirms the increased focus on ostriches’ welfare but shows that further studies are required to ensure the welfare of this species.

## 1. Introduction

The ostrich (*Struthio camelus*), a member of the ratite family, is recently considered an important commercial species and has received interest in farming [1,2] for the production of meat, eggs, feathers, and leather. Especially, there is a reasonable level of demand for ostrich meat [3] due to its nutritive value [3]. In fact, consumers look for healthier alternatives to traditional red meats. This growing interest in ostrich farming has led to an increasing demand for information on breeding practices and their management [4,5,6,7]. Indeed, even though ostrich farming is relatively recent, its rapid growth has not been supported by robust fundamental basic research, like it was instead developed for the industry of the other major livestock species. This rapid development could pose a risk of unethical practices [8] given that the animal welfare requirements would not be met in the absence of any clear and transparent legislation.

There are many definitions of animal welfare [9]—oriented and open to multidisciplinarity—and how it should be assessed regarding different areas of the study [10]. High standards of husbandry such as the care of animals, good housing, protection from the environment, maintaining good health, preventing diseases, recognizing and treating diseases, providing good nutrition, and good stockpersonship should be applied to ensure the good welfare of food-producing animals including ostriches.

With the aim of better understanding how ostriches’ welfare has been studied in the scientific literature and highlighting any gaps, the main topics most associated with animal welfare terms in ostriches over the last forty years (1983–2023) were assessed. Several studies have applied the text mining (TM) and topic analysis (TA) approach to examine the animal welfare of other different species [10,11,12,13,14,15,16]. Using TM, it is possible to extract precious information from textual data (i.e., word incidence and distribution, pattern connection, and predictive analysis) not obtainable with other data analysis approaches.

## 2. Materials and Methods

### 2.1. Literature Investigation and Descriptive Analysis

Relevant studies on ostriches’ welfare published between January 1983 and December 2023 containing at least an English abstract were identified using the Scopus^®^ database (i.e., Elsevier^©^’s abstracts and citation archive).

The Scopus database was chosen due to its accessibility and comprehensive coverage of the scientific literature [17]. The search was conducted on 10 April 2024. Publication date (1983 to 2023), type of article (review and scientific article), language (English), and availability of the abstract were considered as criteria for refining the search. It used the keywords: “Ostriches and behavioual”, “Ostriches and management”, “Ostriches and stress”, “Ostriches and welfare”, “Ostriches and well-being”, “Ostriches and human relationship”, “Ostriches and emotional state”, and “Ostriches and nutrition”. A database created in Microsoft Office Excel^®^ was generated from Scopus^®^ to include all the published documents. This Excel database reported relevant information for each publication, such as authors’ names and affiliations, abstract, time of publication, type of document (e.g., article or review), font of publication (e.g., journal title), and issue. Records were then selected and those that had no abstract, no author name, and that were not articles and reviews were not included. Duplicates were eliminated automatically. In order to assess each document’s eligibility for inclusion and to reduce potential bias, the authors conducted an additional screening, focusing on the specific topic and the species considered. The research group responsible for the initial screening of the documents included a specialist in animal welfare. The inclusion/exclusion criteria chosen are reported in Figure 1. Records associated with other species (e.g., poultry, emu, pig, rabbit, and frog) and other topics (e.g., anatomic studies, evolution matter, genetics, and disease) were excluded.

The flowchart (Figure 1) shows each step of the procedure, presenting the number of documents that were reserved for additional evaluation or definitely excluded. Total documents were n. 987.

Preliminarily, in order to generate a comprehensive profile of the scientific dataset, descriptive statistics were managed, fixing on the year of publication and scientific journal. Pivot tables were applied to estimate the document amount for the year and emphasize the important journals that play a substantial role in the topic.

### 2.2. Text Mining

After the selection of documents for analysis, Rstudio for TM in R (Version 1.3.1093, Free Software Foundation, Boston, MA, USA) subsequent to the transfer procedure was utilized. An Excel sheet was ordered with two distinct columns: “doc_id”, containing the sequential numbering of the 122 documents, and “text”, which encompassed the abstracts of the chosen papers for TM analysis. According to Sebastiani [18], the documents were submitted to the following pre-processing steps:(1)Lowercase conversion.(2)Exclusion of uncommon characters (such as “@”, “/” or “*”), punctuations, numbers, and stop words (e.g., “the”, “a”, “and”, “on”, etc.). Also, words strictly connected with the examined topic or usually utilized, such as chick”, “protein”, “body”, “ratite”, “poultry”, “behaviour”, “bird”, “welfare”, “ostrich”, “farm”, “significant”, “group”, “well-being”, “test”, “animal”, were also removed. Extra white spaces resulting from preceding steps were also removed.(3)Tokenization of text (reducing words to their root form) in order to prevent the same term from being counted in different grammatical forms.

After a document–term matrix (DTM) was constructed, we positioned articles in the rows and terms in the columns as reported by Contiero et al. (2019) [12]. A term frequency-inverse document frequency (TF-IDF) approach was put in application to dispense influences to terms based on both their occurrence within an article and predominance in the documents group. This modification facilitated a more comprehensive assessment of the significance of individual words within the document set. Words presenting the highest relevance (TF-IDF > 1) were represented in a histogram, providing a visual representation of their distribution. A word cloud indicating the most significant words was created utilizing the website https://www.wordclouds.com/ (accessed on 14 April 2024). Larger character dimensions show higher TF-IDF values. Relations between the most significant words (TF-IDF > 1) and all the article words in the body of the text were recognized for a correlation level of ≥0.2. The statistical analysis was performed with an R package (Rstudio; Version 1.3.1093, Free Software Foundation, Boston, MA, USA) and utilities from “tm”, “tidyverse”, “SnowballC”, “wordcloud”, “ggplot2”, and “dplyr”.

### 2.3. Topic Analysis

The Latent Dirichlet Allocation (LDA) approach was used to perform topic modeling analysis. LDA is a hierarchical Bayesian procedure [10] for text analysis that identifies thematic topics by examining word co-occurrence within texts. In LDA, the characteristics of topics are defined in terms of the distribution of words, while the structure of texts is modeled in terms of the distribution of latent topics. The LDA approach, which relies on the analysis of observed texts and words, can reveal the basic latent topic structure, thereby enabling the generation of topic distribution for each document and word distribution for each topic. This methodology has been extensively validated and its effectiveness has been demonstrated across a range of text types and domains [19,20]. The analysis was conducted using the LDA function with Gibbs sampling, which was employed from the “topic models” package in R. The visualization of the most prevalent terms for each issue and their corresponding probabilities was performed with the “tidytext” R library. Before commencing the analysis, a decision was made regarding the number of topics that would be used to divide the corpus into sections. As there is an inherent uncertainty surrounding the optimal number of topics, four models were created, with either 4, 5, 6, or 7 topics, with a final, consensus-based choice of the most informative set of topics made by the research team. Upon the completion of the topic selection process, the research team proceeded to assign indicative labels to each of the five identified topics. To facilitate the topic categorization, the cumulative probabilities of the top 10 terms within each issue were computed, and subsequently, the topics were arranged in accordance with this ranking. The data were presented in the form of bar histograms, with each bar showing the chance of a term occurring within a given topic, as quantified by the beta-value coefficient. This approach, which aligns with the methodology proposed by Nalon et al. (2021) [14], enabled the identification of each topic by assigning a unique label.

## 3. Results

### 3.1. Descriptive Statistics

One hundred and twenty-two out of n. 987 abstracts (12.36%) retrieved and downloaded from Scopus^®^ satisfied the screening and admissibility criteria. Conversely, the documents that were excluded focused on other species and/or topics such as anatomic research, evolution matter, genetics, disease, etc. This category accounted for 77.6% (*n* = 671/865) of the excluded articles. Also were excluded duplicates (*n* = 149/865; 17.2%), articles without the abstract (*n* = 2/865; 0.2%), articles with unidentified authors (*n* = 4/865; 0.5%), and articles that did not fall into the category of articles or reviews (*n* = 39/865; 4.5%).

Research articles (n. 102/122; 83.6%) were the common type of retained records followed by reviews (n. 20/122; 16.4%). Figure 2 shows that the proliferation in the amount of papers published per year started in 1994.

The articles included were published in n. 67 scientific journals. “*Animal Science Journal*”, “*Journal of the South African Veterinary Association*”, and “*Tropical Animal Health Production*” were journals publishing five documents about the topic, respectively, 5/122 records (4.10%); “*Animal Welfare*” and “*Applied Animal Behaviour Science*” with n. 6/122 records each (4.92%); and *South African Journal of Animal Science* with n. 12/122 records (9.84%) (Figure 3).

### 3.2. Text Mining

Once the data had been organized and simplified, the authors had 1169 unique terms from the original n. 122 records. The most common words are visually represented in the histogram (Figure 4). Figure 5 uses a tag cloud to display the words, with the font dimension indicating the importance of each word based on its TF-IDF value. Following data pre-processing and scarceness diminution, no. 1169 terms were retained from the initial no. 122 records. Figure 4 shows the most frequent terms (TF-IDF ≥ 1) represented in relation to the TF-IDF weighting system. Figure 5 represents a tag cloud in which the dimension of the font resembles the TF-IDF significance for each term. The terms with the highest TFIDF were “transport” (2.25), “diet” (2.01), “feed” (1.86), “weight” (1.83), “breed” (1.64), “product” (1.59), “incub” (1.57), “human” (1.49), “manag” (1.42), “temperature” (1.4), “disease” (1.35), “growth” (1.31), “anim” (1.29), “week” (1.26), “female” (1.21), “disease” (1.35), level (1.19), “mortal” (1.18), “nutrit” (1.18), “month” (1.18), “produc” (1.17), “concentr” (1.16), “rate” (1.16), “environ” (1.15), “hatch” (1.13), “slaughter” (1.13), “condit” (1.11), “require” (1.1), “control” (1.1), “skin” (1.08), “feather” (1.07), “perform” (1.06), “experi” (1.06), “develop” (1.05), “present” (1.05), “dietari” (1.05), “differ” (1.03), “male” (1.02), “practic” (1.01), and “vaccin” (1.01).

Table 1 displays the associations between the most frequent terms (with TFIDF ≥ 1.5) and the remaining terms within the matrix, including significant correlations (with a grade of correlation ≥ 0.2).

### 3.3. Topic Analysis

Five topics were chosen as ideal, and labels were assigned to each of them. Table 2 illustrates the name of each issue, the number of papers present in each topic, and their first year of publication.

Figure 6 reports the cumulative probabilities (cps) of topics 1 to 5, along with the top 10 words for each topic, categorized from 1 to 5, also ranked by their cumulative probabilities (cp). Topic 3 (“health and management”), topic 5 (“feeding and nutrition”), and topic 1 (“reproduction”) were the topics with the highest amount of documents, represented by n. 34, n. 25, and n. 23 documents, respectively, followed by topic 4 (“egg production”) with n. 22 documents and topic 2 (“welfare during transport”) with n. 18 documents.

The distribution of the articles from 1983 to 2023 within the five topics is shown in Figure 7. A trendline indicated a rise in the number of documents for topics 2 and 5, while it was observed a decrease in the number of documents for topics 1, 3, and 4.

## 4. Discussion

By employing machine learning approaches such as TM and TA, this study extracted comprehensive insights about ostriches’ welfare from a different array of scientific documents published after 1983. Using these approaches, it was possible to assess wide issues of the topics and distinguish detailed areas with gaps in knowledge and understanding. The amount of published articles on ostriches’ welfare has been increasing beginning from 1994, with peaks both in 1998 and 2014. From 1983 to 1994, the string search identified only one article written in English per year. This interest is understandable considering that animal welfare research began to emerge starting in the 1970s [19] and has since garnered significant attention. The opinion regarding animal welfare in the husbandry system has been a key motivator. This growing emphasis also interests the field of ostriches’ husbandry [21,22], even if their domestication is more recent compared to the longstanding farming of other livestock species [23]. This interest in ostriches could be attributed to their diffusion around several Western countries, the high quality of their production [3], and their rusticity and/or their adaptability to different environments [24].

Within the journals with the main published articles, most were African. This is probably attributable to the ostriches living throughout Africa’s semiarid plains and woodlands [25]. However, international journals such as “*Animal Welfare*” and “*Applied Animal Behaviour Science*” have also shown an interest in these animals. It could be due, as abovementioned, to their diffusion, and to their ability to thrive in challenging environments [25]. However, although they are adaptable animals, it is crucial to ensure optimal animal welfare standards to increase the quality of animal products and to meet evolving consumer preferences [26]. Recently, in fact, they appear more willing to spend a higher price to buy animal welfare-friendly products [27].

The first ten words emerging, classified based on their significance and similar in their meaning, emphasize that one of the considerably investigated issues about ostriches’ welfare is the transport of these species, their feeding and management, and human relationships. Between the different terms, it is remarkable that the term “transport” emerges with higher frequency in TM analysis, as well as within the topics revealed by TA, which also has an increasing trend.

Ostriches are typically transported to a distant processing plant for slaughter, posing a significant challenge to their welfare [28,29]. According to several authors [30,31], transport, consisting of pre-transport, loading, transport, unloading, and post-transport stages, is a critical moment influencing ostriches’ well-being. Handling and transport can lead to weight loss, mortality, and welfare issues in animals, affecting product quality [30,32,33]. Their unique physical characteristics, including a heavy body mass and high center of gravity, make ostriches more susceptible to injuries during handling and transport compared to other species [31]. Several tools are used to handle ostriches during transport, with hooding being recommended as a safe method for birds over 6 months old. It is emphasized by the European Food Safety Authority (EFSA) and South African Ostrich Business Chamber (SAOBC) that hoods should be removed shortly after application, used only when necessary, and taken off soon after loading [34]. It was shown that hooding may cause stress to ostriches due to its disorienting effect, suggesting the need for further research on its effects as a handling method [31].

The EFSA (2004) [34] suggests luring an ostrich with food into a confined space and guiding it into a small triangular enclosure as a low-stress method for capturing ostriches. They also recommend using a triangular restraint for adult ostriches or a shepherd’s staff, in particular for dealing with aggressive male ostriches in a large enclosure. However, the Standing Committee of the European Convention for the Protection of Animals Kept for Farming Purposes (SCECPAFP) [35] and EFSA [34] have banned the use of hooks for controlling ostriches due to the hazardous and stress-inducing procedures which could cause injuries such as neck and head damage, trachea laceration, or fatal outcomes.

In accordance with transport guidelines, various factors need to be taken into account when handling and transporting ostriches, such as their familiarity with each other when they are mixed. For example, the National Animal Welfare Advisory Committee (NAWAC) [36] and EFSA [34] recommend maintaining social groups of animals in the spaces where they are gathered and held, as even minor alterations in their social dynamics can lead to stress-related issues and aggressive behavior. Ostriches are known to be active during the day [37]. During daytime transport, they prefer to stay standing [29], while when transported during the night they tend to sit down and are more relaxed [34,35,36,37,38,39,40] or when they are tired [38]. The optimal duration for withholding feed before transport and slaughter is not determined, as general recommendations may not be suitable for these animals. Further investigation is necessary to establish the optimal duration for withholding feed and water during this phase. Improving handlers’ understanding of their characteristics and needs can enhance both handler safety and bird welfare, reducing transport-related losses.

Furthermore, research is essential to pinpoint the reliable indicators of stress-related behavior, which can help in identifying distressed birds before welfare issues or significant losses arise. Determining the ideal bird density within the trailer is crucial to mitigate losses caused by overcrowding, necessitating further research in this area [31].

About the associations between the words, several observations may be performed. The words “breed” are frequently connected to “season” and “female” probably due to the relationship between the reproductive period and the amount of eggs laid by each female per season, average egg weight, length of laying period, and clutch sequence [41]. The predominant focus on females could be explained because they ovulate spontaneously and the presence of the male is not necessary for production [42]. In ostrich farming, the breeding season is determined by daylight and weather conditions, lasting 6-8 months or more. Ostriches breed seasonally, with peaks in late winter–early spring and early summer, typically from May/June to January/February in the southern hemisphere. They mate in the free-range system or pair/trios with the females reaching maturity at 18–24 months [42]. The average age for a female to lay their first egg is 733 days [43]. Egg production varies greatly among females, with larger eggs produced at the end of the breeding season [42].

Related to ostriches’ welfare, it is interesting to emphasize the impact of diet on their conditioning and, consequently, on their well-being in the husbandry system. The studies on diet focus on their concentration and content as well as the conversion of feed into products. Nutrition plays an important role in ostrich farming, with 75% of the production costs attributed to it [44]. Fertility issues in ostriches are often linked to inadequate breeder diets, emphasizing the importance of protein and energy intake [45]. Ostriches fed high-energy diets have the highest feed conversion ratios, and cholesterol levels in ostriches increase with higher fat and lower protein diets, particularly in those fed lower-energy diets [46].

This investigation highlighted the main welfare-focused research in ostrich farming. The trending topics that have the maximum number of documents are strongly associated, and it is easy to recognize their connection. The most important was the topic of “health and management” (topic 3). The records included within this topic demonstrate a significant scientific interest in managing ostriches and improving their welfare, and consequently their health, within husbandry systems. Within this topic are also studies focusing on behaviors that indicate comfort in ostriches, despite limited research on these animals. These behaviors are suggested as positive welfare markers such as dustbathing and grooming [47]. By dustbathing, ostriches remove ectoparasites and surplus fat. Other research [48] suggested that the playful running and dancing comportment of young ostriches may decrease as they grow older. Finally, ostriches that receive substantial human care from an early age show greater friendliness to humans and willingness to engage with them later in life with respect to ostriches raised under standard commercial practices [49]. In addition, with respect to those reared in the absence of human or foster imprinting, ostriches imprinted by humans have higher survival rates up to four weeks of age [47].

The other most significant topic was “feeding and nutrition” (topic 5). As above, to ensure the healthy development and breeding success of ostriches, it is crucial to provide them with a tailored diet that meets their specific nutritional needs. While some have mistakenly believed that poultry diets could suffice for ostriches [50], it is essential to recognize that these birds have distinct vitamin and mineral requirements [51].

The following most important topics are correlated and are “reproduction“ (topic 1) and “egg production” (topic 4). It is important to highlight that stocking density does not influence egg production, hatchability, age of sexual maturity, or duration of egg production season, but a space of more than 300 m^2^ determines a decrease in egg weight and day-old chick weight. Conversely, an area less than 100 m^2^ determined a reduction in fertility rate [52].

Despite being one of the most frequent words, transport is the last topic with only 18 articles related to animal welfare during this phase; however, its trend is increasing as well as topics related to “feeding and nutrition”.

It is important to underline the limitations of the procedure used in this analysis. Firstly, the search strings may not have embraced all the potential synonyms, restricting the range of documents selected. Documents outside of the Scopus^®^ database were not included, which could have reduced the completeness of the review. The search parameters, including the presence of abstracts in the English language and detailed screening criteria, may have influenced the amount of documents examined. Additionally, the review approach focused on titles and abstracts rather than a comprehensive evaluation of each record. Nevertheless, this review offers precious insights into ostriches’ welfare, emphasizing key topics and clarity gaps.

## 5. Conclusions

The analysis reveals that the research on ostriches’ welfare predominantly addresses transport and breeding management, with a significant emphasis on negative welfare indicators. Conversely, issues regarding the welfare of ostriches during production and the assessment of welfare-positive indicators were scarcely present.

Overall, ostriches’ welfare is a relatively new, multidisciplinary field—as defined by the various topics recognized—that is still developing. This highlights the need for further research in this emerging field to expand the knowledge base and further advance in this topic.

## Figures and Tables

**Figure 1 vetsci-11-00477-f001:**
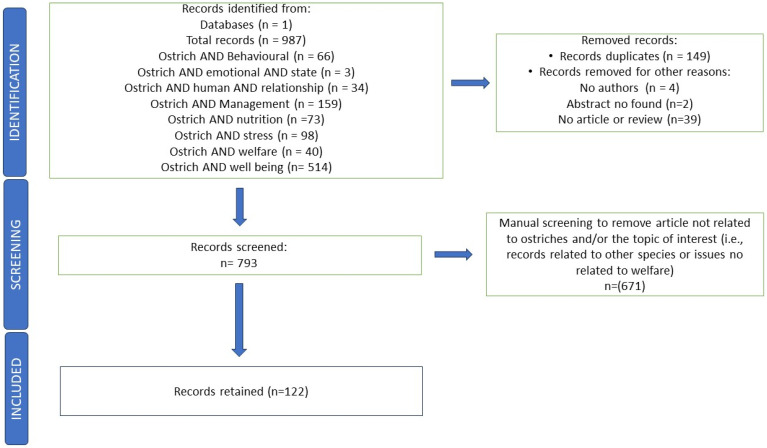
Flowchart reports the total amount of documents included and/or excluded in the review process regarding ostriches’ welfare.

**Figure 2 vetsci-11-00477-f002:**
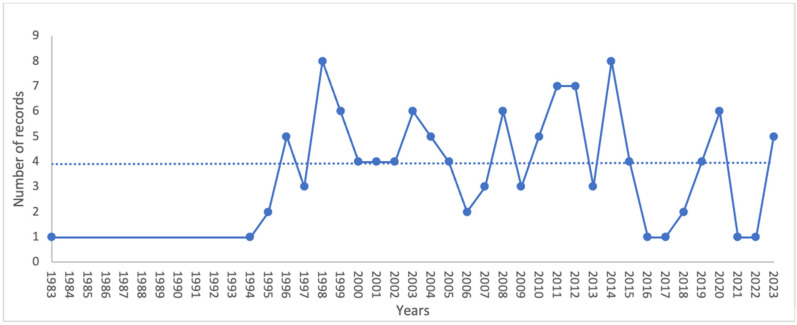
Number of records on ostriches welfare distributed by publication year (1983–2023) within the 122 records selected for inclusion in the review.

**Figure 3 vetsci-11-00477-f003:**
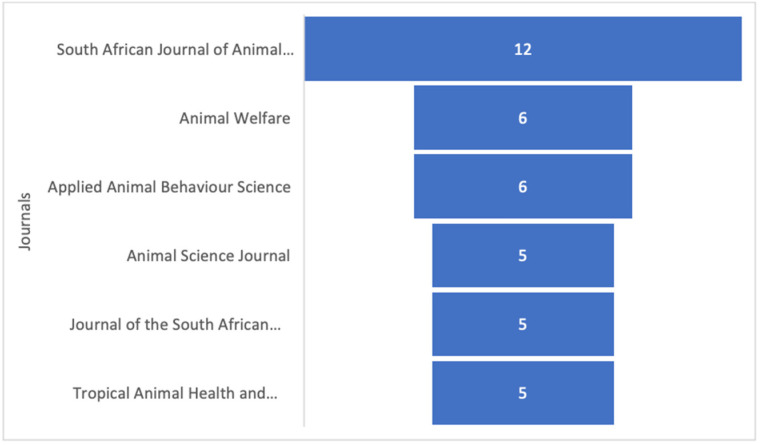
Main scientific journals on the topic.

**Figure 4 vetsci-11-00477-f004:**
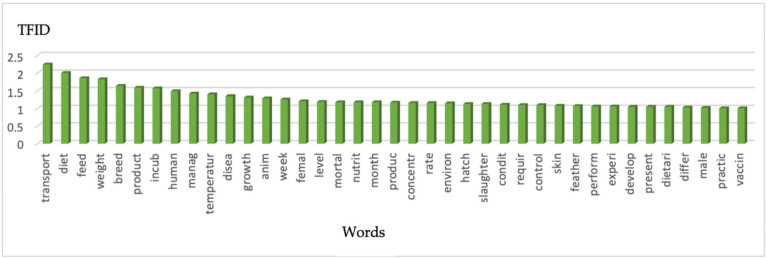
The histograms report the most relevant words (TFIDF ≥ 1) detected from no. 122 documents examined.

**Figure 5 vetsci-11-00477-f005:**
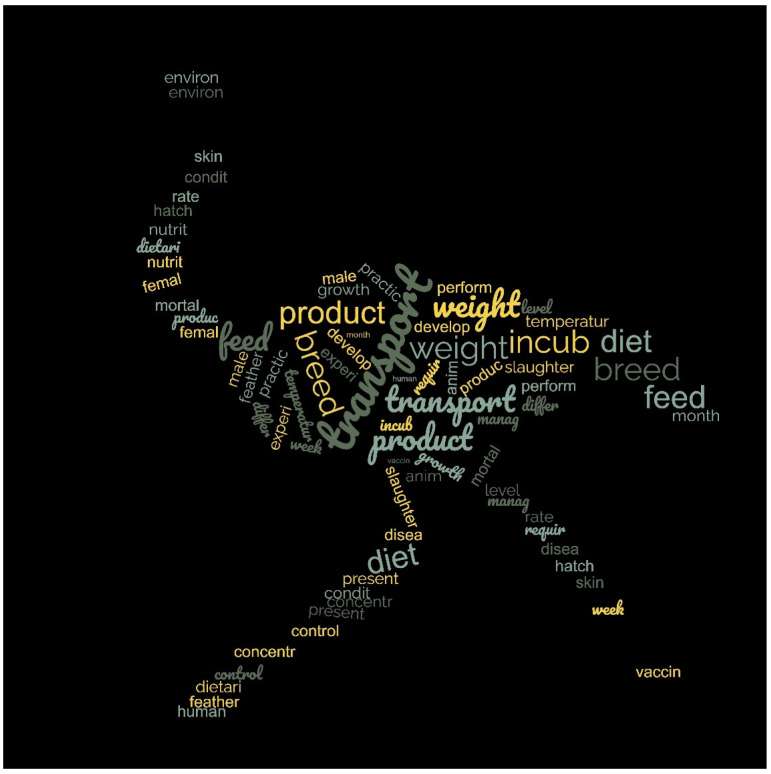
Tag cloud indicating the most significant words identified in the 122 examined documents. The size of each word is proportional to its significance.

**Figure 6 vetsci-11-00477-f006:**
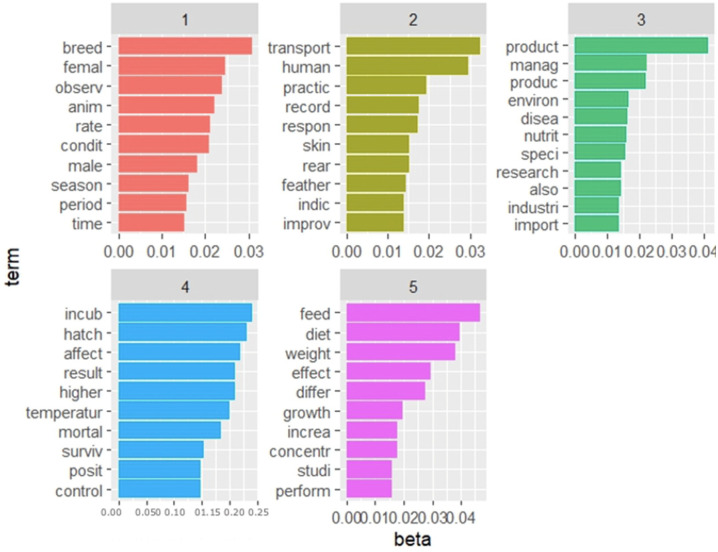
Histograms representing the most relevant words for the five main topics present in the LDA.

**Figure 7 vetsci-11-00477-f007:**
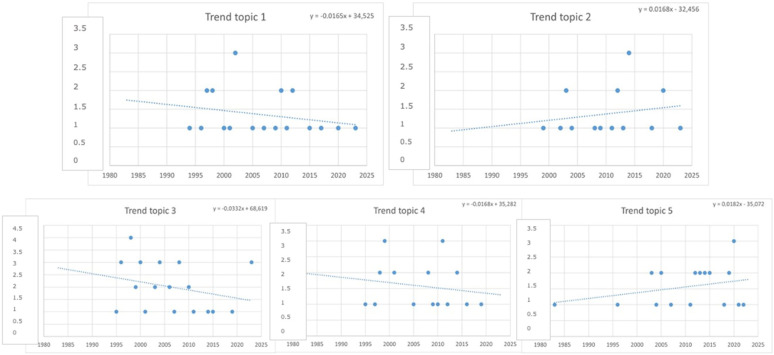
Trend in the five topics (1983–2023).

**Table 1 vetsci-11-00477-t001:** Association between the most frequent words (with TFIDF ≥ 1.5) and the remaining words within the matrix for correlation grade (r) ≥ 0.2.

Words (TF-IDF ≥ 1.5)	Related Words (Grade of Correlation ≥ 0.2)
**Breed**	Season (0.80); female (0.53); metabolism (0.51); fewer (0.47); easili (0.46); entir (0.44); earlier (0.42); pair (0.41)
**Diet**	Contain (0.66); finish (0.58); starter (0.58); cholesterol (0.55); grower (0.51); prestart (0.51); gain (0.50); concentr (0.49); formul (0.48); haematolog (0.47); phase (0.46); manufactur (0.44)
**Feed**	Convert (0.46); convers (0.44); sole (0.43)
**Incub**	Embryo (0.71); bottom (0.66); batch (0.64); middl (0.61); back (0.59); embryon (0.54); storag (0.52); front (0.51); exceed (0.49); impair (0.48); store (0.46); forc (0.45); hatch (0.44); nest (0.44); vertic (0.44); immedi (0.43); posit (0.43); prior(0.41); shell (0.41)
**Product**	Produc (0.52); industry (0.44); system (0.41)
**Transport**	Vehicl (0.60); america (0.50); stress (0.49); preliminari (0.48); state (0.48); posttransport (0.47); lost (0.46); ship (0.46); unit (0.46); identifi (0.45); literatur (0.45); road (0.42); stressor (0.42); phosphokinase (0.41); research (0.41)
**Weight**	Gain (0.52); bodi (0.51); factori (0.47); anova (0.46); grower (0.46); starter (0.43); length (0.41); diurnal (0.40)

**Table 2 vetsci-11-00477-t002:** Name of each topic and the number of papers present with the relative first year of publication.

Topic Number	Label of Topic	Papers (n)/From Year
1	Reproduction	23/1994
2	Welfare during transport	18/1999
3	Health and management	34/1995
4	Egg production	22/1995
5	Feeding and nutrition	25/1983

## Data Availability

For other information contact the corresponding author (michela.pugliese@unime.it).

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
