# Peer review of "Text Mining and Topic Analysis for Ostriches’ Welfare Based on Systematic Literature Review from 1983 to 2023"

_vetsci, 2024, doi:10.3390/vetsci11100477_

Round 1

Reviewer 1 Report

Comments and Suggestions for Authors

The ostriches welfare is relatively new and multidisciplinary field. This review is interesting for many people.

1.     Figure 4 adds ordinate headings and units

2.     What does Figure 5 mean? What are the essential differences from Table 1 and Figure 4?

3.     Does Figure 6 have units in the horizontal and vertical coordinates? Many of the keywords in Figure 6 are missing letters, causing many words to be incomplete and unable to identify.

4.     Some words in Table 1 and Figure 5 are also missing letters.

5.     As we all know, expressing natural behaviors and expressing positive emotions are very important to animal welfare. However, the key content or keywords analyzed in this paper do not include behaviors and emotions. Is it because they were not included in statistics or because there are too few research literatures?

Author Response

Dear Reviewer,

Thank you very much for your time and all your comments.

 We thank you for your precise and thoughtful comments and constructive criticism, which has led to a better manuscript.

We revised the manuscript concerning the suggestions and more detailed answers are given below. 

The changes made in the manuscript to address comments are written in red.

  1. Figure 4 adds ordinate headings and units.
  2. The figure has been modified as suggested.

  1. What does Figure 5 mean? What are the essential differences from Table 1 and Figure 4?
  2. Figure 5 represents a tag cloud in which the dimension of font resembles the TF-IDF significance for each term. Table 1 show the association between most frequent words (with TFIDF ≥1.5) and the remaining words within the matrix for correlation grade (r) ≥ 0.2, while the figure 4 reports the most relevant words (TFIDF ≥1) detected of no. 122 documents examined as histograms.
  3. Does Figure 6 have units in the horizontal and vertical coordinates? Many of the keywords in Figure 6 are missing letters, causing many words to be incomplete and unable to identify.
  4. Coordinates of figure 6 are horizontal. Words in figure 6 present missing letters, given that are submitted to tokenization process (reducing words to their root form) in order to prevent the same term from being counted in different grammatical forms.

  1. Some words in Table 1 and Figure 5 are also missing letters.
  2. See above.

  1. As we all know, expressing natural behaviors and expressing positive emotions are very important to animal welfare. However, the key content or keywords analyzed in this paper do not include behaviors and emotions. Is it because they were not included in statistics or because there are too few research literatures?
  2. The analysis reported show that the research on ostriches’ welfare predominantly addresses transport and breeding management, with a significant scarceness on other indicators of welfare as natural behaviors and expressing positive emotions. Therefore, they are not included in the statistical analysis because there are too few research literatures.

On behalf of the Authors

Prof. Michela Pugliese

Reviewer 2 Report

Comments and Suggestions for Authors

GENERAL COMMENTS: the authors submitted a manuscript regarding a systematic review of ostriches' welfare using text mining and topic analysis. This is an excellent topic due to the lack of studies in this field. Also, the authors contributed by discussing the next steps of new research. I suggested some improvements and some doubts to be clarified as described below:

SIMPLE SUMMARY: this part finished with "(...) 987 publications". Considering that a simple summary is directed at the general audience, I suggest stating the main result of your research.

MATERIAL AND METHODS: regarding the systematic review, did you consider inserting any assessment of the risk of bias that may affect the cumulative evidence (e.g., publication bias, selective reporting within studies)? If not, this is a critical analysis in a systematic review.

Lines 72-73: the Microsoft software is not necessary.

Line 79: the indication of authors is also unnecessary. This is the reason to have the topic "Conceptualization".

Line 84: Please state the inclusion/exclusion criteria.

Line 104: remove Excel mention.

RESULTS: Line 167: why did you include reviews? Most of the systematic reviews only included articles.

DISCUSSION: Lines 269-270: the production efficiency is not linked with animal welfare. AW refers to life quality and not production quality. Please consider changing this statement.

Lines 274-275: this atends the industry, not the animals. This part of your discussion is too much directed to animal production. Welfare is more than this. Please change this sentence.

Line 285: idem

Comments on the Quality of English Language

Please review the English. Minor improvements are necessary.

Author Response

Dear Reviewer,

Thank you very much for your time and all your comments.

 We thank you for your precise and thoughtful comments and constructive criticism, which has led to a better manuscript.

We revised the manuscript concerning the suggestions and more detailed answers are given below. 

The changes made in the manuscript to address comments are written in red.

  1. GENERAL COMMENTS: the authors submitted a manuscript regarding a systematic review of ostriches' welfare using text mining and topic analysis. This is an excellent topic due to the lack of studies in this field. Also, the authors contributed by discussing the next steps of new research. I suggested some improvements and some doubts to be clarified as described below:

SIMPLE SUMMARY: this part finished with "(...) 987 publications". Considering that a simple summary is directed at the general audience, I suggest stating the main result of your research.

  1. The sentence has been modified (lines 17-18).
  2. MATERIAL AND METHODS: regarding the systematic review, did you consider inserting any assessment of the risk of bias that may affect the cumulative evidence (e.g., publication bias, selective reporting within studies)? If not, this is a critical analysis in a systematic review.
  3. The information has been included as suggested (line 13).
  4. Lines 72-73: the Microsoft software is not necessary.
  5. Done.
  6. Line 79: the indication of authors is also unnecessary. This is the reason to have the topic "Conceptualization".
  7. Done.
  8. Line 84: Please state the inclusion/exclusion criteria.
  9. The inclusion/exclusion crietria is summarized in Figure 1 and detailed in lines 74-81.
  10. Line 104: remove Excel mention.
  11. Done.
  12. RESULTS: Line 167: why did you include reviews? Most of the systematic reviews only included articles.
  13. While it is correct that many reviews only include articles, it is necessary to note that this choice is not mandatory, but rather a discretionary decision made by the authors. We preferred to include reviews in line with other papers that used this methodology. Given that TM and TA methodologies allow the conversion of qualitative data into quantitative data, incorporating reviews can offer valuable insight using these analytical tools.
  14. DISCUSSION: Lines 269-270: the production efficiency is not linked with animal welfare. AW refers to life quality and not production quality. Please consider changing this statement.
  15. The statement has been modified.
  16. Lines 274-275: this atends the industry, not the animals. This part of your discussion is too much directed to animal production. Welfare is more than this. Please change this sentence.
  17. The statement has been modified.
  18. Line 285: idem
  19. The statement has been modified.

On behalf of the Authors

Prof. Michela Pugliese

Reviewer 3 Report

Comments and Suggestions for Authors

This article uses TM and TA techniques and uses Scopus as the data base to analyze 987 publications retrieved.

The analysis process is appropriate and the results presented are well discussed. In general, there is no big problem and it can be published with just a few minor revises.

There are only a few questions need to be clarified.

1.      Line13: toprovideï¼›  a space is missing.  Should be “ to provide”

2.      Line 223: Were chosen five topics----ï¼›

I’m not sure what this sentence meant.  Could it be: Five topics were chosen----

3.      Line 243: ---fort topics 1,3 and 4.    Does it means: for topics 1,3 and4

4.      When the “number of articles” is expressed, there is “n.” before the number. Is this necessary? If so, it should be unified, because in some places “no.” is used.

5.      Captions for tables should be placed above the table while captions for figures are placed below the figure.  Captions for table 1 and table 2 were misplaced.

Author Response

Thank you very much for your time and all your comments.

 We thank you for your precise and thoughtful comments and constructive criticism, which has led to a better manuscript.

We revised the manuscript concerning the suggestions and more detailed answers are given below. 

The changes made in the manuscript to address comments are written in red.

  1. This article uses TM and TA techniques and uses Scopus as the data base to analyze 987 publications retrieved.

The analysis process is appropriate and the results presented are well discussed. In general, there is no big problem and it can be published with just a few minor revises.

There are only a few questions need to be clarified.

Line13: toprovideï¼›  a space is missing.  Should be “ to provide”

  1. Done.
  2. Line 223: Were chosen five topics----ï¼›

I’m not sure what this sentence meant.  Could it be: Five topics were chosen----.

  1. Thank you for the suggestion. The sentence has been modified.
  2. Line 243: ---fort topics 1,3 and 4.    Does it means: for topics 1,3 and4.
  3. Done.
  4. When the “number of articles” is expressed, there is “n.” before the number. Is this necessary? If so, it should be unified, because in some places “no.” is used.
  5. The “n” has been unified in whole document.
  6.       Captions for tables should be placed above the table while captions for figures are placed below the figure.  Captions for table 1 and table 2 were misplaced.
  7. As suggested, captions for table 1 and table 2 has been placed above the tables.

On behalf of the Authors

Prof. Michela Pugliese

Round 2

Reviewer 2 Report

Comments and Suggestions for Authors

The authors made a suitable improvement to the manuscript. However, some missed issues must be changed, as suggested in the first review round, as below:

- Line 84: Please state the inclusion/exclusion criteria of the animal welfare specialist (remove the APa abbreviation).

- In a systematic review, assessing the risk of bias that may affect the cumulative evidence (e.g., publication bias, selective reporting within studies) is mandatory. Please include which analysis was performed to give this information.

Author Response

Dear Reviewer,

Thank you very much for your time and all your comments.

We revised the manuscript concerning the suggestions and more detailed answers are given below. 

The changes made in the manuscript to address comments are written in red.

  1. Line 84: Please state the inclusion/exclusion criteria of the animal welfare specialist (remove the APa abbreviation).
  2. The sentence has been rephrased and the abbreviation has been removed. The information is included in the lines above (lines 78-80) to understand the reader better.
  3. In a systematic review, assessing the risk of bias that may affect the cumulative evidence (e.g., publication bias, selective reporting within studies) is mandatory. Please include which analysis was performed to give this information
  4. To minimize the impact of publication bias, reviewers make efforts to recognize all significant documents relevant for inclusion in systematic reviews. New methodologies such as text mining are considered useful for performing systematic reviews more quickly (thus meeting exact policy and practice timescales and increasing their cost efficiency), minimizing the impact of publication bias, and reducing the possibility that relevant documents are missed (O’Mara-Eves et al. Systematic Reviews 2015, 4:5). In the present systematic review, to minimize the bias, has been applied a double approach that included both a TM and a manual screening performed by the Authors (see lines 76-82). This approach is also applied in other studies that used this method (e.g. Masebo et al., 2023.; Chen et al., 2021; McMillan et al., 2023; Nalon et al., 2021).

On behalf of the Authors

Prof. Michela Pugliese
